# Preparation and Properties of a Novel Cross-Linked Network Waterborne Polyurethane for Wood Lacquer

**DOI:** 10.3390/polym15092193

**Published:** 2023-05-05

**Authors:** Yuanyuan Zhou, Yan Liu, Binjie Xin, Ying Qin, Guankun Kuang

**Affiliations:** 1College of Chemistry and Chemical Engineering, Shanghai University of Engineering Science, Shanghai 201620, China; 2College of Textile and Fashion Engineering, Shanghai University of Engineering Science, Shanghai 201620, China

**Keywords:** water-based wood lacquer, polycarbonate diol, cross-linked network, microphase separation, stain resistance

## Abstract

Waterborne polyurethane (WPU) is a waterborne coating with excellent physicochemical properties. Its deficiencies of water resistance, chemical resistance, staining, and hardness have limited the wide application of polyurethane in the wood lacquer market. In this study, polycarbonate diols (PCDL) were used as soft segments and WPCU was modified by cross-linking using Trimethylolpropane (TMP) to prepare polycarbonate type WPU (WPCU) with cross-linked network structure. The new wood lacquer was prepared by adding various additives and tested by applying it on wood board. The successful synthesis of WPCU was determined by FTIR testing, and the cross-linking degree of WPCU was probed by low-field NMR. The viscosity of the cross-linked WPCU emulsion showed a decreasing trend compared to the uncross-linked WPCU emulsion, and WPCU-2 had the smallest particle size. Compared with the uncrosslinked WPCU film, the crosslinked WPCU film had lower water absorption (2.2%), higher water contact angle (72.7°), excellent tensile strength (44.02 MPa), higher thermomechanical, and better water and alcohol resistance. The effect of crosslinker content on the microphase separation of WPCU chain segments on the surface roughness of the film was investigated by SEM. The wood paint prepared by WPCU emulsion has good dry heat resistance, chemical resistance, and adhesion, and the hardness of the wood paint when the TMP content is 3% reaches H. It also has good resistance to sticky stains, which can be used to develop new wood lacquer.

## 1. Introduction

With the development of society and the economy, the requirement for environmental protection in the chemical industry is higher and higher. Low volatile organic compounds (VOCs), water-based coatings are getting more and more attention [1,2,3,4,5]. Waterborne polyurethane (WPU) is a waterborne coating with good compatibility, low-temperature resistance, flexibility, good adhesion, and film-forming [6,7]. WPU has been widely used in coatings, inks, and adhesives [8,9,10]. However, due to the lower molecular mass and many hydrophilic groups on its chain segments, WPU has poor water resistance, thermal stability, and mechanical properties [11,12]. Ordinary WPU cannot be applied in some stringent conditions in wood, automobiles, ships, etc., developing an advanced WPU with excellent water resistance and mechanical properties is crucial.

Polyurethane is usually composed of polyols (soft segments), diisocyanates (hard segments), and low molecular weight diols (chain extenders, hard segments) and solvents [13,14]. Traditional polyols mainly include polyester polyols and polyether polyols [15]. Polycarbonate diol (PCDL) is the raw material for synthesizing a new generation of polycarbonate polyurethane. Several studies have shown that polycarbonate-based polyurethane (PCU) has superior properties to conventional polyurethane. W. Kuran et al. [16] compared PCDL-based and polyester-based PU, affirming PCU’s water resistance and higher mechanical properties. The reason is that the acid group cannot be produced when the carbonate group hydrolyzes. García-pacios et al. [17] synthesized WPU emulsion with polyether and polycarbonate (1000 Da) as substrates and coated it on 304 stainless steel, which has high adhesion strength, ethanol resistance, and high gloss performance. Wu et al. [18] compared WPU prepared by polyethylene glycol (PCL and PBA), polyether ethylene glycol (PTMG), and polycarbonate glycol (PCDL), and found that PCDL-based WPU had high hydrolysis resistance and hardness. Lee et al. [19] showed that polycarbonate functional groups have high polar levels and strong intermolecular forces and that PCU has stronger mechanical properties, anti-hydrolysis, and anti-oxidation. The above team started with the soft segment component and developed a new polymer to solve the application problem.

For PCDL-based waterborne polyurethane (WPCU), the WPU prepared by PCDL with different molecular weights has been studied [20]. Lee et al. [21] designed WPU from PCDL with a molecular weight of 2000, 1000, and 500, respectively. The WPU prepared from PCDL with a molecular weight of 2000 had the best mechanical properties. The team [22] then studied the performance changes of different types of PCDL-based WPU with a molecular weight of 2000. García-pacios et al. [23] studied WPU made from PCDL (500–3000 Da) with different molecular weights. Gao et al. [24] indicated that polyurethane had a higher self-healing ability when the molecular weight of PCDL was 2000. The effects of hard segment content [25], solid content [26], and hydrophilic chain extender content [27] on the properties of WPCU were also studied.

To obtain higher-performance PCDL-based polyurethane, researchers modified WPCU to expand its use. UV-curable waterborne polyurethane acrylate composite resin was prepared by Zhen et al. [28], combining the scratch and abrasion resistance tensile strength of the polyurethane with the optical properties, weather fastness, and adhesion of the acrylate. The hyperbranched WPCU prepared by Wang et al. [29] has better thermal properties as tensile strength increased by 1.9 times (28.15 MPa) and and elongation at break increased by 1.5 times (543.8%) compared with the linear WPCU. Yang et al. [30] synthesized Polymethylhydrosiloxane modified PCU to produce promising biostable polyurethane materials. Fang et al. [31] investigated the effect of polymethyl methacrylate (PA) content on composite elasticity. Wang et al. [32] extended the antibacterial prospect of WPCU emulsion with a quaternary ammonium salt. These modification methods do not involve cross-linking modification, which can make polyurethane gain a cross-linking structure to improve the performance of WPCU.

Trimethylolpropane (TMP) is a polyol with three hydroxyl groups, which can react with three isocyanates to obtain a cross-linked structure in WPU. TMP has been widely used to prepare WPU [33,34,35]. Xu et al. [15] modified WPU with TMP. The results showed that the glass transition temperature (21 °C) of cross-linked WPU film was higher than that of WPU film (10 °C). Liu et al. [14] used TMP as a cross-linking agent to prepare high-elastic WPU with an elongation of 2100%. It’s Young’s modulus is 35 MPa. Zhu et al. [36] considered that the adhesion and water resistance of cross-linked WPU was better than that of linear WPU. Only two references mention WPCU preparation with TMP. Wang et al. [37] studied the influence of PBI/PCDL molar ratio on WPU performance but did not involve studying TMP content on WPCU. Meng et al. [38] prepared cross-linked WPU by mixing polycarbonate glycol/polycaprolactone glycol (PCDL/PCL). Studies show that the WPU emulsion has good thermal stability when the trimethylolpropane (TMP) content is 2.5%, and the molar ratio of PCDL/PCL is 2:1. But this team did not examine the effect of TMP content on the mechanical properties of WPCU. Few teams have studied the effect of TMP on WPCU in-depth, and there is a lack of research on the impact of the ratio of chain extender to the cross-linking agent on WPCU. Furthermore, the effect of the cross-linking structure on the membrane surface roughness and microphase separation should be investigated.

Despite the excellent performance of WPCU with a cross-linking structure, few people have investigated the application of WPCU on wood lacquer. Wood lacquer is a kind of resin lacquer used in wood products, polyester, polyurethane lacquer, etc., and can be divided into water and oil. Oil paint has relatively high hardness and good fullness, but water-based paint is environmentally protected. Water-based wood paint mainly includes four types: (a) water-based wood paint with acrylic acid as the main component, mainly characterized by good adhesion but poor hardness and chemical resistance; (b) wood paint with acrylic acid and polyurethane synthesis as the main component, improve the chemical resistance of acrylic ester paint and hardness can reach h; (c) polyurethane water-based paint, its comprehensive performance, high fullness, hardness can reach H~2H; (d) pseudo-water paint, which need to add curing agent and chemicals, high solvent content, great harm to the human body. Waterborne polyurethane resin contains a large number of carbamate bonds.

Wood lacquer should have the following characteristics: can smooth the surface of wood products; avoid substrate scratched by hard objects; prevent moisture penetration; avoid dry cracking caused by direct sunlight. Wood with water-based coating for the resin itself is more demanding, according to the specific application field, and needs to have a certain hardness, gloss, and chemical resistance, to facilitate transportation and storage. Still, it also needs a certain resistance to return viscosity. At the same time, how to make the WPU wood paint have more similar solvent resistance, stain resistance, adhesion, and hardness to the solvent-based wood paint has become an urgent problem for researchers to solve. Based on these, selecting the resin skeleton becomes a key first step. Furthermore, the molecular weight should be large enough to maximize the chain entanglement. Ultimately, cross-linking the membrane will also help improve the strength, wear resistance, weather resistance, and stain resistance.

In conclusion, applying WPCU with the cross-linked structure to the preparation of wood paint is of great significance. In this paper, the modification of TMP as a cross-linking agent produces a novel WPCU emulsion that can improve the thermal stability, water resistance, alcohol resistance, hardness, adhesion, and mechanical properties of WPCU membrane and study the surface morphology of the WPCU membrane. A new water-based wood lacquer was developed based on the blending of a new modified WPCU dispersion and functional additives. The performance of wood paint was evaluated by analyzing dry heat resistance, chemical resistance, stain resistance, hardness, and adhesion.

## 2. Experimental

### 2.1. Materials

Isophorone diisocyanate (IPDI, 98 wt% purity) and 1,4-Butanediol (BDO) (analytical grade) were provided by China Aladdin Chemical Co., Ltd. (Shanghai, China). The Dimethylolpropionic acid (DMPA, 98 wt% purity), Trimethylolpropane (TMP, 95.5 wt% purity), Triethylamine (TEA, 98 wt% purity), Ethylenediamine (EDA, 90 wt% purity) and Acetone (ACE, 99.5 wt% purity) were purchased from Sinopharm Chemical Reagent Co., Ltd. The PCDL (Mn = 2000 g/mol) was provided by Jining Huakai Resin Co., Ltd. (Shandong, China). Bismuth Catalysts (DY-20) were supplied by Shanghai Kaiyin Chemical Co., Ltd. (Shanghai, China). Defoamer (H-286), Wetting agent (H-1400) and leveling agent (H-112), and dispersant (H-788) were purchased from Guangzhou Hengsike New Material Co., Ltd. (Guangzhou, China). Thickener (HEUR-1) was supplied by Shanghai Guben Industrial Co., Ltd. (Shanghai, China). Deionized water made at the laboratory. Molecular sieves were added to dry BDO and acetone before use.

### 2.2. Preparation of WPCU Emulsions

Table 1 displays the formulation of the WPCU. At the beginning of the experiment, PCDL was added to a three-mouth round-bottom flask and then immersed to the oil bath. The flask was then vacuumed out for two hours at 100 °C and used to exclude water from the PCDL. After that, the temperature reduced to 80 °C. Then add the IPDI and stir for tow hours in the presence of DY-20. The DMPA was then added to introduce the -COOH. The reaction continued over the next 1 h at 80 °C.

After the system was cooled to 40 °C, acetone was added to adjust the viscosity. Subsequently, BDO was added as the chain amplicon, heating the system to 60 °C and reacting for two hours. Afterward, TMP was used as a cross-linking agent to respond with the unreacted NCO groups at 60 °C for one hour. Eventually, the TEA was added to react with -COOH for 30 min at 35 °C. The solid content of the waterborne polyurethane emulsion was fixed at 35%. After the reaction, the mass of the deionized water was calculated from the total mass of the reagent. Then, the quantitative deionized water and defoamer were added to the prepolymer slowly at high-speed agitation (1000 rpm) under an ice-water bath. Eventually, the addition of EDA as a posterior chain amplification agent completely reacts with the remaining -NCO in the system. The reaction process and the cross-linking mechanism of WPCU are shown in Figure 1.

The TMP contents of the WPCU were 0%, 1%, 2%, 3% and 4%, corresponding the BDO contents of the WPCU were 7%, 6%, 5%, 4% and 3%, respectively recorded as WPCU-0, WPCU-1, WCPU-2, WPCU-3 and WPCU-4.

### 2.3. Preparation of WPCU Film

A WPCU film was prepared by injecting 15 g of the emulsion into polytetrafluoroethylene plates and left at room temperature for seven days. Subsequently, the polyurethane films were tested after 24 h in a vacuum oven at 60 °C.

### 2.4. Preparation of WPCU Wood Lacquer

The modified WPCU emulsion made by 94 wt% was stirred at a certain rotational speed, then 1–2 wt% wetting agent, 0.2–0.4 wt% leveling agent, and 1–2 wt% dispersant were added successively to adjust the viscosity of the thickener to the proper value, stir well to get the varnish.

### 2.5. Coating of Wood Lacquer

2 mL of WPCU emulsion was applied to a primed wood board (purchased from Zhanchen Coating Group Co., Ltd. (Shanghai, China)) with a 150 um wire rod and then dried at 30 °C to obtain a coating of wood lacquer.

### 2.6. Methods of Characterization

#### 2.6.1. The Characterization of Emulsion

##### Appearance and Stability Measurement of WPCU Emulsions

The stability of the WPCU emulsion was tested by a desktop centrifuge (TDL-80-2B, Shanghai Anting Scientific Instrument Factory (Shanghai, China)) to rotate the aqueous dispersion at 2000 rpm for 20 min. After that, if the aqueous dispersion does not precipitate, the storage stability of the aqueous dispersion will last for six months; otherwise, it will be less than six months. We also observed the emulsion morphology six months later. The stability test was performed according to the requirements of GB/T6753.3-1986 (China), where the emulsion was stored at room temperature for 6 months, and the emulsion was observed for delamination and precipitation. The performance of transparency, color, and uniformity of the dispersion was regarded by visual inspection [38].

##### Particle Size Analysis

A drop or two of WPCU emulsion was added to a beaker containing 100 mL of distilled water. The particle size of the emulsion was measured by dynamic light scattering laser particle size analyzer (DLS, (Nano-ZS90, Malvern, UK)). Each sample was tested three times, and then the median was taken.

##### Viscosity Measurement

Viscosity measurements were performed using a rotating viscometer (NDS-5S, Shanghai Changji Geological Instrument Co., Ltd. (Shanghai, China)). Each sample was tested three times, and then the median was taken.

#### 2.6.2. The Characterization of WPCU Film

##### Hardness Measurement

Pencil hardness: The emulsion was applied on a tinplate sheet, dried at room temperature, and placed in an oven at 60 °C for 24 h. According to the requirements of GB/T 6739–2006 (China), the hardness of the WPCU film was tested using a 750 g loading pencil hardness tester [39].

##### Anti-Water and Anti-Alcohol Measurement

Both water and alcohol resistances were determined according to GB/T 9274-88 (China) [28]. The WPCU-coated tinplate sheet was immersed in water, and the coating changes were observed after 16 days to judge the water resistance of the coating. The WPCU-coated tinplate sheet was immersed in 50% ethanol solution, and the coating changes were observed after two hours to judge the alcohol resistance of the coating.

##### Water Absorption Measurement

The WPCU membrane was cut into small squares of 1 cm × 1 cm and soaked in distilled water for 48 h. Then dried with filter paper, and then the water absorption rate was calculated as follows:$$W=\frac{m_{2}-m_{1}}{m_{1}}\times 100\%$$
where W is the water absorption rate of the film, and m_1_ and m_2_ are the quality of the dried film before and after immersion in the sample, respectively. Each sample was tested three times, and then the median was taken.

##### Structure Characterization

The sample structure was identified using a Fourier Transform infrared spectroscopy (Perkin-Elmer Inc., (Waltham, MA, USA)).

##### Crosslinking Degree Analysis

Low field nuclear magnetic resonance analyzer (LF-NMR (PQ001, Niumai Analytical Instruments Co., Ltd., (Suzhou, China))) was used to test the crosslinking degree of WPCU films. The magnetic field intensity was 0.517 T, the frequency was 15 MHz, the pulse width of 90° and 180° was 2.8 μs and 5.8 μs, respectively, the waiting time was 2 s, the number of accumulation was 8, and the test temperature was 90 °C. In NMR relaxation, the attenuation of magnetization with time (t) during lateral relaxation can be expressed by the following theoretical equation [40]:y=Aexp−tT2−12qM2t2+Bexp−tT2+Cexp−tT2s+A0
where A, B and C respectively represent the proportion of the signals of the cross-linked chain, the pendant chain and the free chain in the whole signal. A_0_ is a fitting parameter of no practical significance. T_2_ and T_2s_ are the corresponding transverse relaxation time. qM_2_ is the residual dipole moment above the glass transition temperature.

##### Thermal Gravimetric Analysis

The thermal stability of the WPCU film was studied using a thermogravimetric analyzer (TGA-4000, Perkin Elmer Inc., USA). The heating rate was 10 °C/min, the sample mass was 5 to 9 mg, the temperature ranged from 30 °C to 600 °C, and the nitrogen atmosphere was used.

##### Static Contact Angle Analysis

The WPCU film was tested by a contact angle instrument (SDC-200S, Dongwan Shengding Precision Instrument Co., Ltd., (Dongwan, China)). Each sample was tested five times at different locations, and the median was taken.

##### SEM Measurement

The surface morphology of the WPCU film was observed using a scanning electron microscope (SEM, (Hitachi, su800, Tokyo, Japan)). The test was performed in a low vacuum environment and at a 3 kv acceleration voltage.

##### Mechanical Performance

The WPCU film was tensile tested by an electronic universal testing machine (AGS-X 10KN STD, Shimadzu, Japan). The dumbbell-shaped samples were cut from 1-mm-thick WPCU membranes in 50 mm × 10 mm × 4 mm. All samples were stretched at 30 mm/min and tested five times per sample.

#### 2.6.3. The Characterization of Wood Lacquer

The appearance of the wood paint coating was observed by visual inspection. The gloss of the wood paint was measured by a 60° gloss tester (CS300, Color Spectrum Technology Co., Ltd. (Hangzhou, China)). The adhesion of the coating was tested according to GB/T 9286-1998 by using the lacquer scribe (QFH-A, Shenzhen Wright Instruments & Equipment Co., Ltd. (Shenzhen, China)) for the 100-grid method. The wooden boards coated with different WPCU were put into the oven at 70 °C for 15 min, to judge the dry heat resistance of the coating by observing the state of the coating. The alkali resistance test was performed at room temperature according to GB/T 4893.1-2005. The test solution is 5 g/L NaHCO_3_, the test time is 1 h, and placed for 1 h to observe. The test and evaluation method to acid resistance test is the same as alkali resistance. The test solution is 5% H_2_SO_4_ solution, the test time is 2 h, and the surface of the coating is observed after 1 h. The test and evaluation method of alcohol resistance test is the same as alkali resistance, the test solution is 50% ethanol solution volume, the test time is 1 h, and the test observation is made after 1 h. The evaluation method of stain resistance test is the same as alkali resistance, using Pearl River brand pure black ink as the test solution, the test time is 5 h, and the test observation is conducted after 1 h of placement.

## 3. Results and Discussion

### 3.1. Characterization of Emulsion

#### 3.1.1. Storage Stability of the Emulsion

Figure 2 shows the appearance of the WPCU emulsion containing different TMP and BDO content. As can be seen from the picture, WPCU-0 and WPCU-1 emulsions show a blue transparent appearance. However, the turbidity of WPCU increases with the increase of TMP content. WPCU-2, WPCU-3, and WPCU-4 appear milky white with light blue color. As shown in Table 2, all WPCU emulsions showed no delamination or precipitation after centrifugation, indicating that all emulsions were stable. After six months of emulsion storage, none of the emulsions showed solidification, yellowing or precipitation, which corresponded to the centrifugation results and affirmed the stability of the emulsions.

#### 3.1.2. Particle Size and Viscosity of the Emulsion

The effect of TMP content on the particle size distribution and viscosity of WPCU emulsion is shown in Figure 3, respectively. It can be seen that the average particle size of WPCU-0 emulsion to WPCU-2 emulsion showed a decreasing trend with the increase of TMP content and the decrease of BDO, which may be due to the fact that the addition of too much chain extender will make the molecular chains of prepolymer too long, thus leading to the easy entanglement of molecular chains in the emulsification process and making it difficult to disperse the prepolymer. However, with the increase of TMP content, the average functionality of the system increases, the tightness of intermolecular entanglement increases, leading to decreases the space volume, and the PU prepolymer gradually develops from a linear molecular to a cross-linked structure. The excessive cross-linked structure makes it difficult to disperse the WPCU-3 prepolymer in water, which leads to an increase in particle size. WPCU-4 was added with less chain extender, but the excessive cross-linked structure restricts the migration of hydrophilic groups to the surface, so it had a moderate particle size. So proper cross-linking and chain extension neither limit the migration of the hydrophilic groups nor cause entanglement due to excessively long segments. As TMP content increases and BDO content decreases, viscosity decreases from 349 mPa·s to 34.5 mPa·s. Too much chain extender will make the prepolymer more difficult to disperse in water, resulting in higher system viscosity. After introducing TMP, the tightness of intermolecular entanglement increases, increasing the emulsion’s viscosity, so the WPCU-1 has the most significant viscosity. However, with the further reduction of BDO content, even with the increase of TMP content, the system’s viscosity showed a downward trend, indicating that the content of the chain extender had a more significant impact on the viscosity of the system.

### 3.2. Characterization of WPCU Film

#### 3.2.1. FTIR Spectroscopy Analyses

The FTIR spectra of the WPCU film are shown in Figure 4. The spectra of WPCU-0 to WPCU-4 were very similar. The peaks at 3333 cm^−1^ were attributed to the tensile vibration of N-H of amide bonds [41]. The absorption peaks near 2860~2940 cm^−1^ were attributed to the alkane groups’ C-H stretching vibration [42]. Furthermore, the absorption peak near 1726 cm^−1^ was assigned to the stretching vibration peak of C=O bonds from the urethane groups [43]. A sharp and characteristic peak at 1530 cm^−1^ was the bending vibration absorption peak of the N-H bond in the amide bond. Several peaks at 1060~1255 cm^−1^ are related to the C-O-C tensile vibration in the urethane groups [44]. Moreover, the asymmetric stretching vibration peak of -NCO near 2270 cm^−1^ ceases to be visible for all WPCU, indicating that all -NCO groups are depleted and the termination reaction is complete [45].

#### 3.2.2. NMR Analysis

NMR has been widely used to test the crosslink density of materials [46,47,48,49]. NMR is very sensitive to changes in molecular motion and can be used to detect spin-spin relaxation times of chains [50] and to obtain useful structural information about the network, such as how crosslinked chains differ from free and suspended chains. Three different chain segments are included in WPCUs with a cross-linked structure. The first type consists of cross-linked chains connected to the network at both ends. The second type consists of suspended chains with free ends and rings. The third one is composed of free chains and small molecule additives. These three components have different mobility and exhibit different spin-spin relaxation time measurements in NMR [43]. The cross-linked chains have the lowest mobility and therefore the shortest relaxation times. The faster the proton relaxation curve decays, the higher the cross-link density of the polymer [49]. Attenuation curve of WPCU with different TMP contents are shown in Figure 5. It can be seen that the recession rate gradually increases with the increase of TMP content, indicating that the content of cross-linked chains in the system is increasing and the cross-link density is increasing.

#### 3.2.3. Water Absorption and Contact Angle Analysis

The results of the water absorbance and contact angle tests of the WPCU films are shown in Figure 6. The results show that when the TMP content is less than 2%, the water absorption of the film gradually decreases as the amount of TMP increases. However, when the content of TMP was more than 3%, the water absorption rate of WPCU film increased, which may be due to the negative effects caused by too much cross-linking agent and too little chain extenter. The TMP crosslinker increases the crosslink density of the chain segments and the intermolecular forces increase, leading to an increase in microphase separation, as a result, the WPCU film becomes rougher and the film-forming state deteriorates, making it easier for water molecules to enter [10]. This result corresponds to the findings of SEM tests. When TMP content reaches 1%, the water contact angle of the WPCU film added 13° compared with WPCU-0, indicating that the formation of the cross-linked structure can improve the contact angle of the WPCU film. However, the contact angle did not change much as the TMP content continued to increase; this trend may have resulted from the excessive cross-linking structure, resulting in film formation defects and increased film roughness. According to Wenzel’s equation [51,52]:cos θ*=rcosθ
where θ* refers to the surface (rough surface) contact angle, θ refers to Young’s contact angle (on an ideal surface), and r refers to the roughness ratio. For the hydrophilic surface (θ <90°), the greater the roughness, the smaller θ*, and for the hydrophobic surface (θ <90°), the greater the roughness, the greater θ*. Therefore, the contact angle did not continue to grow under the combined effect of the increase of WPCU film roughness and the expansion of the cross-linked network. All in all, when the content of the hydrophilic chain extender is the same, the water resistance of WPCU films modified with an appropriate amount of TMP is better than those of the unmodified WPCU films.

#### 3.2.4. Anti-Water and Anti-Alcohol Properties of the WPCU Films

The water and alcohol resistance properties of the WPCU-0 to WPCU-4 films are shown in Figure 7. As shown in Figure 7a, When the TMP content was 3%, the coating had the best water resistance. When the TMP content continues to increase and the BDO content continues to decrease, the water resistance of the coating decreases. This trend may have resulted from the film formation defects, and it was easier for water to enter the interior of the layer. In Figure 7b, from the whitening of the coating, when the TMP content was less than 2%, the WPCU film was severely whitened after immersion in 50% ethanol solution for 2 h. WPCU-2 and WPCU-3 showed good alcohol resistance. To sum up, the formation of the cross-linked structure is beneficial to the increase of the water resistance and alcohol resistance of the film.

#### 3.2.5. SEM Analysis

The micrographs of the WPCU films were obtained by SEM, as shown in Figure 8. It would be observed that the surface of the WPCU films gradually became rough, and there are more and more aggregates in the system with the TMP content increasing. This phenomenon could be due to too many cross-linked structures. The soft and hard segments of WPCU are microphase separated, like in Figure 8f, resulting in film formation defects. The soft and hard segment separation scheme of WPCU film dispersions is illustrated in Figure 9. From the perspective of microscopic morphological structure, in WPCU, strong polar and rigid urethane groups and other groups can form hydrogen bonds between molecules due to their sizeable cohesive energy and aggregate together to form a hard segment microphase region. These microphase regions are glassy sub-crystals or crystallites at room temperature. The less polar polyester segments aggregate together to create soft segment domains. Although the soft and hard segments are miscible to a certain extent, the hard and soft segments are thermodynamically incompatible, resulting in microscopic phase separation [53]. The attraction between hard segments is much greater than between soft segments. The TMP cross-linking agent increases the cross-linking density, affecting the molecular chain’s creep properties. Moreover, the intermolecular force increases, resulting in increased microphase separation, thus, rougher film formation [10].

#### 3.2.6. Thermal Stability Analysis

Use TGA to analyze the thermal stability of the polyurethane. Figure 10 demonstrates the TG and DTG curves of WPCU films with distinct TMP contents. The DTG curves showed different degradation stages and the temperature at the maximum degradation rate. The temperature of 5 wt% weight loss (T_5wt%_), the temperature of 10 wt% weight loss (T_10wt%_), and the temperature of maximum degradation (T_dmax_) were summarized in Table 3.

The DTG curve shows that the hard segment degradation is evident in the temperature range of 290 to 320 °C, while the soft segment degradation is in the temperature range of 330 to 400 °C. T_5wt%_ is usually considered the onset decomposition temperature. It was clear that T_5wt%_ increased from 259.51 to 267.85 °C as TMP content rose from 0% to 4%. Furthermore, the temperatures of T_10wt%_ moved from 288.97 to 291.3 °C, and T_dmax_ moved from 368.40 to 378.26 °C as the content of TMP increased from 0% to 4%. An increase in T_5wt%_, T_10wt%_, and T_dmax_ indicated that the rise of TMP content helped improve the thermal stability of WPCU films. Still, the film formation defect occurs when the crosslinker exceeds 3%, and the thermal performance cannot be significantly improved. Introducing TMP’s cross-linked network structure enhanced the inter/intramolecular action. Furthermore, it slows down the decomposition of the macromolecular chain of the WPCU film and improves the thermal stability of the polymer.

#### 3.2.7. Mechanical Properties of WPCU Films

The mechanical properties of the prepared films were evaluated by testing the tensile strength and fracture elongation. As shown in Figure 11, all prepared membranes have yield points. The results indicated that with the increase of TMP, the tensile strength and fracture elongation of WPCU film are significantly improved. The detailed results are shown in Table 4. After adding the TMP crosslinker, the tensile strength of the WPCU membrane was increased from 14.20 MPa to 44.02 MPa. In terms of elongation at break, with the increase of TMP content, the elongation at break of WPCU film gradually decreases because segment fluidity decreases with the gradual increase of cross-linking density. As shown in Figure 11b, compared with WPU modified by 3-aminopropyltriethoxysilane (APTES) [22], itaconic acid-based cross-linking (IHA) with UV-curable [54], hydroxylated melamine in solid form (MOH-S) [55], mixing isobornyl acrylate (IBOA) with UV-curable [11], aminoethyls aminopropyl dimethicone (AEAPS) [6], and (3-(2-Aminoethyl)aminopropyl) trimethoxysilane (AEAPTMS) [7], TMP cross-linking agent WPCU film has better mechanical properties. Moreover, compared with TMP-modified WPU, TMP-modified WPCU also had excellent tensile strength, which was increased from the maximum 40.3 MPa reported in the literature to 44.02 MPa [56,57].

### 3.3. Characterization of Wood Lacquer

As seen from Table 5, adding some TMP to WPCU to increase the internal cross-linking can improve the coating gloss, possibly because the cross-linking structure or the penetrating network structure makes the coating structure fuller, which is conducive to the surface leveling. However, the excessive cross-linking agent may cause the coating to become rough and reduce the gloss of the coating. All the coatings have good adhesion to level 1 to meet the needs of wood use. All coatings have good dry heat resistance and no cracking or bubbling on the surface after 15 min at 70 °C. The hardness of WPCU film increases from B to H with the increase of TMP content. It shows that the formation of a crosslinked structure will increase the hardness of the film. In wood lacquer, the hardness of the lacquer film is one of the key performance indicators to determine the performance of a lacquer, the hardness of the product film is directly proportional to the scratch resistance and wear resistance index, the higher the hardness, the better the scratch resistance and the higher the wear resistance index [58]. From the viewpoint of hardness, WPCU wood lacquer coating at 3% TMP addition can reach H, which can meet the needs of wood lacquer. As seen from Figure 12, when the content of the crosslinker is less than 2%, the coating has no bulge or severe swelling phenomenon after the chemical resistance test, but the stain resistance is poor. When the content of crosslinker is 2% and 3%, the coating has a good chemical resistance and stain resistance; when the content of crosslinker reaches 4%, and the content of BDO is 3%, the chemical resistance is not abnormal, but the coating has poor stain resistance.

## 4. Conclusions

In summary, we have developed a new polycarbonate-based waterborne polyurethane with a cross-linking structure to prepare a water-based wood lacquer with good chemical resistance, dry heat resistance, stain resistance, hardness and adhesion. The formation of the cross-linked structure improved the thermal stability, water resistance, ethanol resistance, and mechanical properties of WPCU (44.02 MPa). It reduced the water absorption of the WPCU membrane for 24 h (2.2%). SEM analysis shows that an excess of TMP roughens the surface of the WPCU membrane because the increased cross-linking density leads to enhanced microphase separation. When the TMP content is more than 3%, the coating has good hardness (H), and WPCU-3 has good stain resistance. The modification of WPCU by TMP cross-linking has strong operability and practical value, which can provide a useful reference for the modification of waterborne wood lacquer.

## Figures and Tables

**Figure 1 polymers-15-02193-f001:**
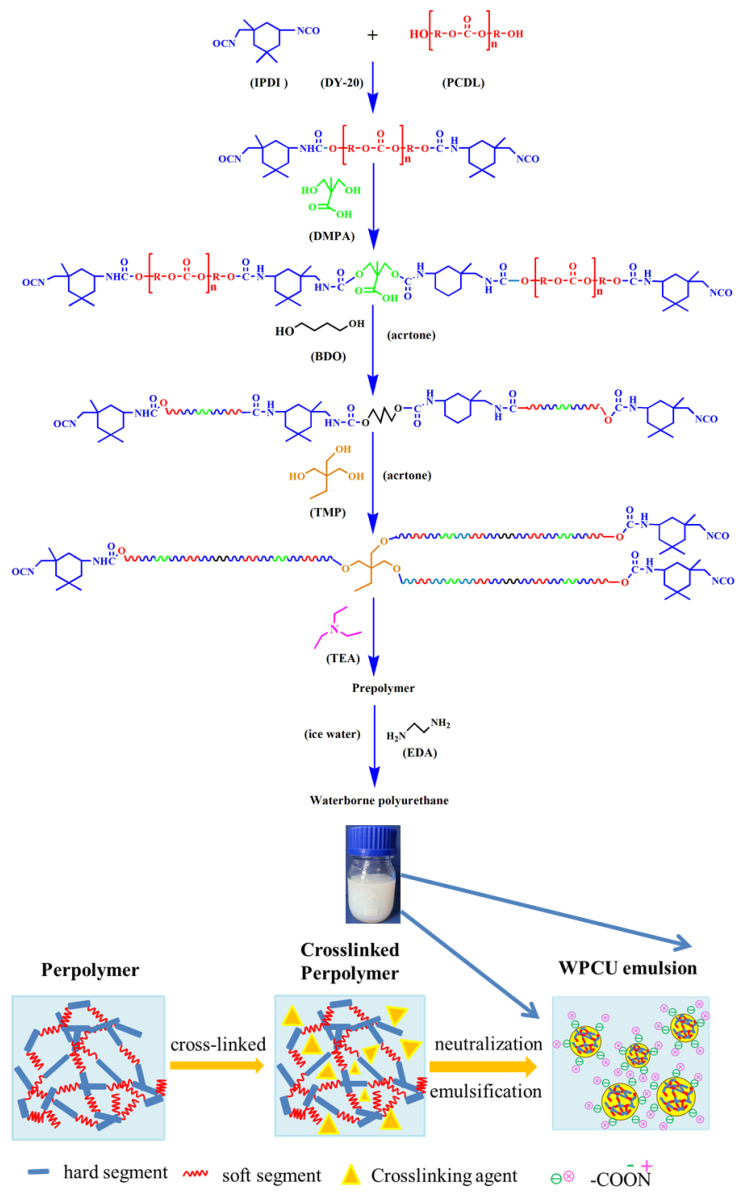
Schematic illustration of the preparation of Cross-linked WPCU emulsion.

**Figure 2 polymers-15-02193-f002:**
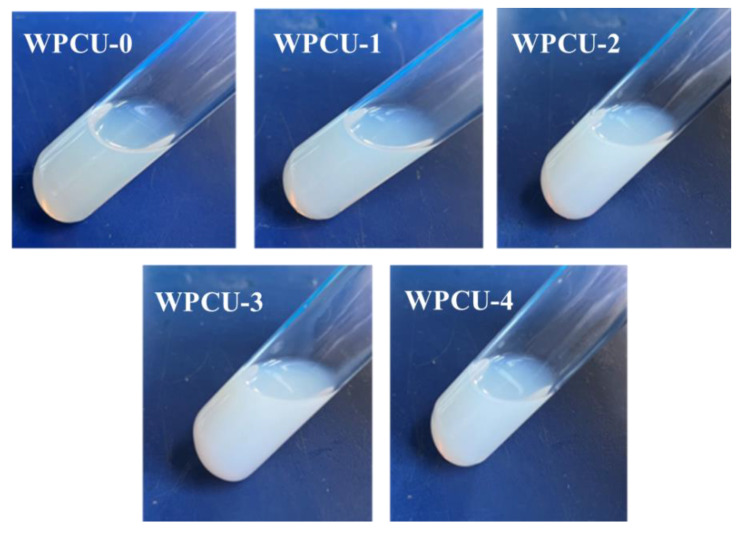
The digital photos of WPCU emulsions.

**Figure 3 polymers-15-02193-f003:**
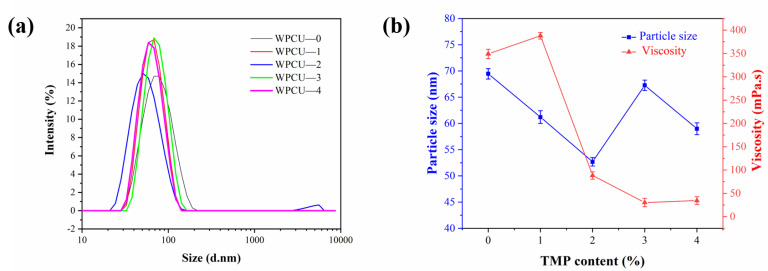
(**a**) Particle size distributions of WPCU emulsions; (**b**) Particle size and viscosity of WPCU emulsion with different TMP contents.

**Figure 4 polymers-15-02193-f004:**
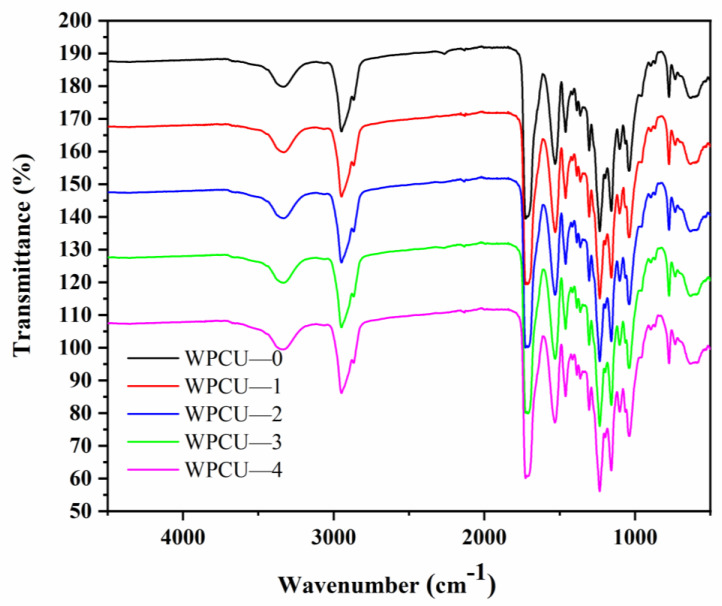
FTIR spectra of WPCU with different TMP contents.

**Figure 5 polymers-15-02193-f005:**
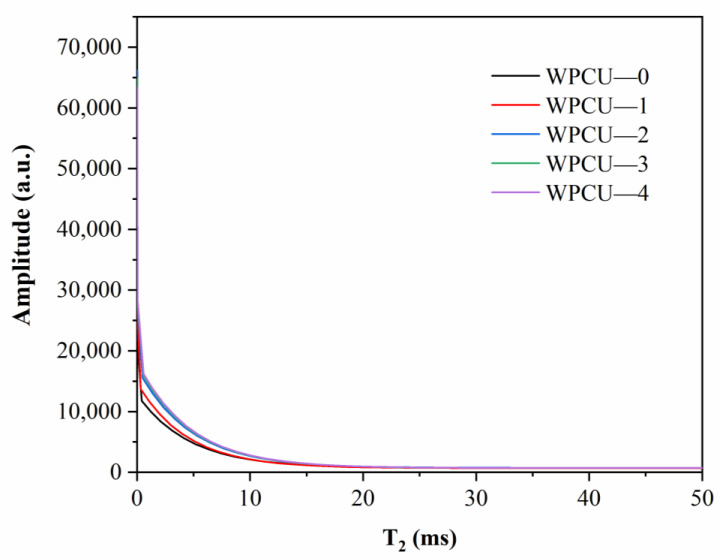
Attenuation curve of WPCU with different TMP contents.

**Figure 6 polymers-15-02193-f006:**
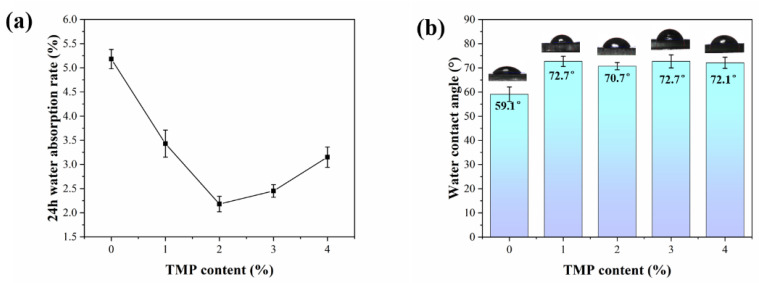
(**a**) Water absorption property and (**b**) water contact angles of the WPCU films.

**Figure 7 polymers-15-02193-f007:**
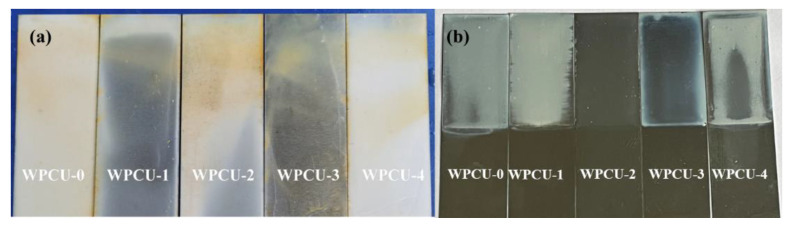
(**a**) Anti-water and (**b**) Anti-alcohol properties of the WPCU films.

**Figure 8 polymers-15-02193-f008:**
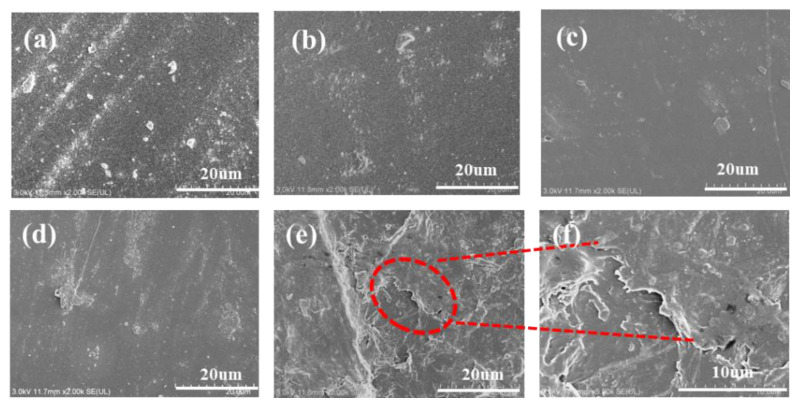
Micrographs of the WPCU films, (**a**) WPCU-0; (**b**) WPCU-1; (**c**) WPCU-2; (**d**) WPCU-3; (**e**) WPCU-4 at 20 um size, and (**f**) WPCU-4 at 10 um size.

**Figure 9 polymers-15-02193-f009:**
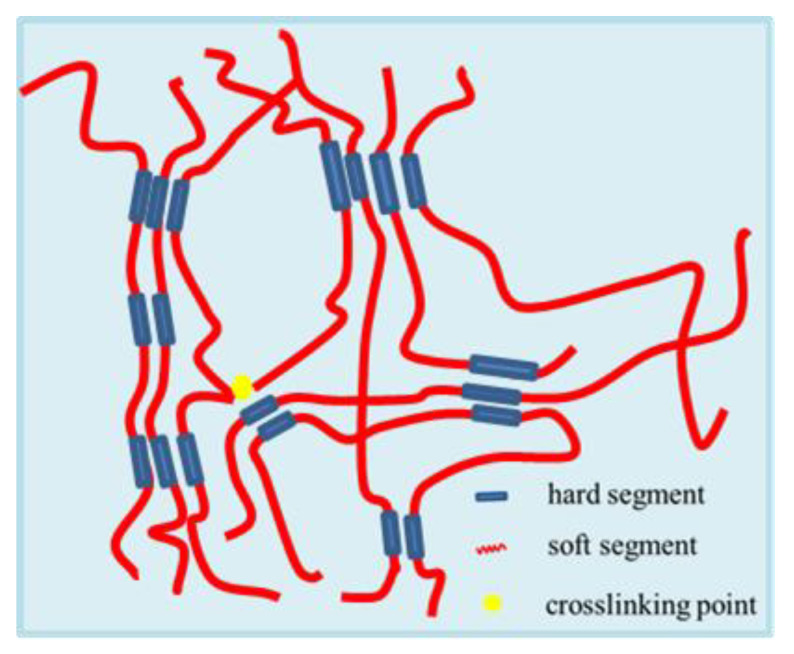
Schematic illustration of the separation of soft and hard segments of WPCU film.

**Figure 10 polymers-15-02193-f010:**
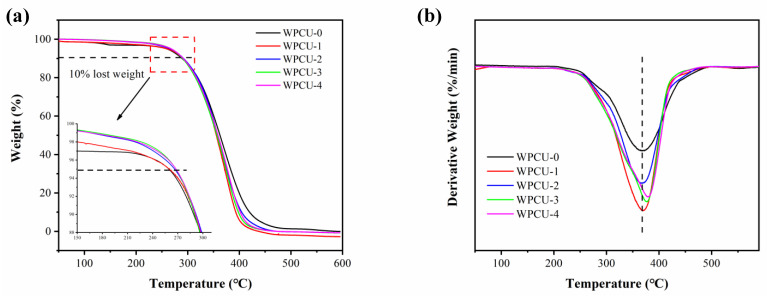
(**a**) TG and (**b**) DTG curves of WPCU films.

**Figure 11 polymers-15-02193-f011:**
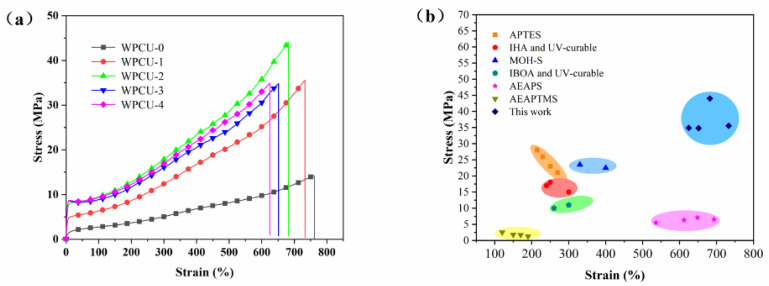
(**a**) Tensile-strain curves of WPCU films and (**b**) comparison of mechanical properties with reported papers.

**Figure 12 polymers-15-02193-f012:**
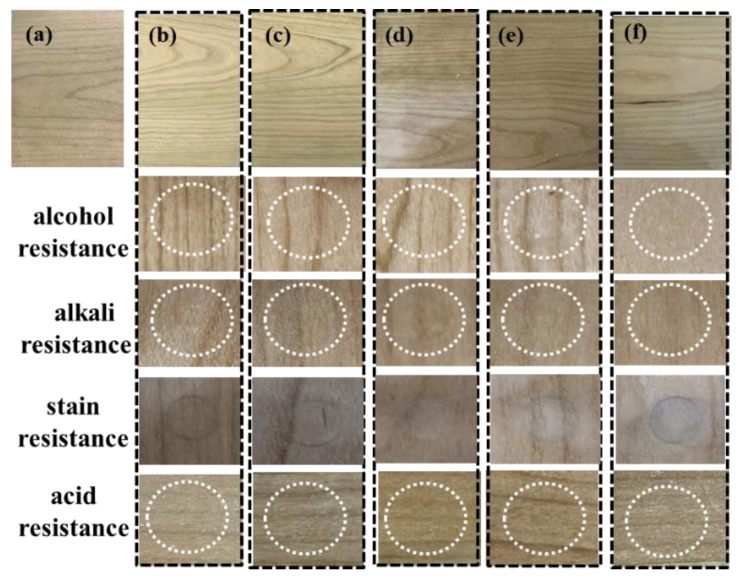
(**a**) Wood surface without wood lacquer; (**b**–**f**) Surface and performance test of wooden boards coated with WPCU-0, WPCU-1, WPCU-2, WPCU-3, and WPCU-4 wood lacquer.

**Table 1 polymers-15-02193-t001:** Amount of the reagents used for preparing WPCU.

Sample	IPDI (mol)	PCDL (g)	DMPA (mol)	TMP (mol)	BDO (mol)	TEA (mol)	Total mass (g)	Water (g)
WPCU-0	0.079	26.11	0.013	0	0.040	0.013	50.98	94.68
WPCU-1	0.079	26.11	0.013	0.004	0.034	0.013	50.98	94.68
WPCU-2	0.079	26.11	0.013	0.008	0.027	0.013	50.98	94.68
WPCU-3	0.079	26.11	0.013	0.011	0.023	0.013	50.98	94.68
WPCU-4	0.079	26.11	0.013	0.015	0.017	0.013	50.98	94.68

**Table 2 polymers-15-02193-t002:** The appearance and stability of WPCU emulsions.

Sample	Appearance	Centrifugal Stability	Stability (Month)
WPCU-0	Blue translucent	Stable	>6
WPCU-1	Blue translucent	Stable	>6
WPCU-2	Milky white with light blue	Stable	>6
WPCU-3	Milky white with light blue	Stable	>6
WPCU-4	Milky white with light blue	Stable	>6

**Table 3 polymers-15-02193-t003:** Thermal properties of WPCU films.

Sample	T_5wt%_ (°C)	T_10wt%_ (°C)	T_dmax_ (°C)
WPCU-0	259.51	288.97	368.40
WPCU-1	260.21	290.65	370.72
WPCU-2	266.19	291.89	373.83
WPCU-3	267.75	290.05	377.34
WPCU-4	267.85	291.35	378.26

**Table 4 polymers-15-02193-t004:** Tensile properties of WPCU films.

Sample	Breaking Strength (MPa)	Breaking Elongation (%)
WPCU-0	14.20	759.37
WPCU-1	35.54	732.67
WPCU-2	44.02	681.92
WPCU-3	34.81	651.47
WPCU-4	34.89	624.79

**Table 5 polymers-15-02193-t005:** Gloss, adhesion, and resistance to the dry heat of wood lacquer.

Sample	Gloss (%)	Adhesion (Level)	Hardness	Resistance to Dry Heat
Wood board	1.5	/	/	/
WPCU-0	87.2	1	B	pass through
WPCU-1	86.8	1	HB	pass through
WPCU-2	89.4	1	H	pass through
WPCU-3	91.5	1	H	pass through
WPCU-4	90.3	1	H	pass through

## Data Availability

No new data were created.

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
