# Peer review of "Preparation and Properties of a Novel Cross-Linked Network Waterborne Polyurethane for Wood Lacquer"

_polymers, 2023, doi:10.3390/polym15092193_

Round 1

Reviewer 1 Report

The paper is interesting and the authors have done a lot of experiments. However, some revisions are needed:

First of all, please check carefully the manuscript as they are wrong formats, inconsistent style, and spacing problem

Please rewrite the abstract, it is not informative enough

The spacing of keywords, please check

The style of citation in text is not correct, it should be in bracket [1].

The title of section 2.4 and 2.5 is the same, please check and amend.

Please rewrite section 2.6.3 in a more proper way.

“As shown in Table 2, the average particle size of all WPCU emulsions was less than 100 nm”, particle size was not shown in Table 2, please check.

What do you mean by the “stability” under the column “Centrifugal stability” in Table 2?

Section 3.1.2 “It ' s concluded that the mean particle size of the WPCU emulsion showed a trend of decrease at the beginning and then increased with higher TMP content.” It is possible that the particle size at 3% TMP content is an outlier?

Section 3.2.1 – so no different in chemical structure within WPCU-0 to WPCU-4?

Section 3.2.2 – “indicating that the formation of the cross-linked structure can improve the contact angle of the WPCU film” where do you prove that there are cross-linked structure?

More in-depth discussions are needed. Please explain why 3% TMP content and 4% BOD content can lead to the best wood lacquer?

There are no references at all in the discussion section, please revise.

Conclusion – “Considering the practical application and cost control, the TMP content is 3%, and the BDO content is 4%, as the best wood lacquer formula.” Can you please revise this sentence  

Reviewer 2 Report

Comments

The authors prepare novel cross-linked network waterborne polyurethane for wood lacquer. The similar previous studies have been investigated.

1. Please compare to the properties of other studies, which use similar raw materials.

https://onlinelibrary.wiley.com/doi/full/10.1002/app.52364

https://www.sciencedirect.com/science/article/pii/S0300944019305739

https://www.tandfonline.com/doi/abs/10.1080/01932691.2021.1956526

2. How do the authors check the crosslinking degree of materials?

3. Figure 6 is not clear.

4. What is the number of experimental repetitions?

5. Please conduct statistical analysis.

6. Please correct grammatical and typographical errors.

Round 2

Reviewer 1 Report

The paper has been improved and now in a better form. However, some improvement are still needed:

1. you mentioned that the WPCU is a novel system, please highlight the novelty of the system.

2. Section 3.2.1 and other section - no references

3. Section 3.2.2 - this part is not written in a scientific style, please rewrite and provide more in-depth discussion, with references.

Overall, the discussion is lacking and has to be improved.

"However, when the content of TMP was more than 3%, the water absorption rate of WPCU film increased, which may be due to the negative effects caused by too much cross-linking agent and too little chain extenter" can you elaborate more on this statement? how too much cross-linking affect the WA rate?

The whole manuscript has to be revised to make sure it is written in a more concise manner.

In conclusion, adding some future perspectives would be beneficial.
